# Hawkmoths evaluate scenting flowers with the tip of their proboscis

Alexander Haverkamp[1†], Felipe Yon[2†], Ian W Keesey[1], Christine Mißbach[1], Christopher Koenig[1], Bill S Hansson[1], Ian T Baldwin[2], Markus Knaden[1‡], Danny Kessler[2*‡]

[1]Department of Evolutionary Neuroethology, Max-Planck Institute for Chemical Ecology, Jena, Germany; [2]Department of Molecular Ecology, Max-Planck Institute for Chemical Ecology, Jena, Germany

**Abstract** Pollination by insects is essential to many ecosystems. Previously, we have shown that floral scent is important to mediate pollen transfer between plants (*Kessler et al., 2015*). Yet, the mechanisms by which pollinators evaluate volatiles of single flowers remained unclear. Here, *Nicotiana attenuata* plants, in which floral volatiles have been genetically silenced and its hawkmoth pollinator, *Manduca sexta*, were used in semi-natural tent and wind-tunnel assays to explore the function of floral scent. We found that floral scent functions to increase the fitness of individual flowers not only by increasing detectability but also by enhancing the pollinator's foraging efforts. Combining proboscis choice tests with neurophysiological, anatomical and molecular analyses we show that this effect is governed by newly discovered olfactory neurons on the tip of the moth's proboscis. With the tip of their tongue, pollinators assess the advertisement of individual flowers, an ability essential for maintaining this important ecosystem service.

*For correspondence: dkessler@ice.mpg.de

[†]These authors contributed equally to this work
[‡]These authors also contributed equally to this work

## Introduction

Floral scent has been associated with insect pollination since the 18[th] century (*Sprengler, 1793*); however, the complex functions of floral volatiles have been only recently investigated in more detail, due to the availability of new molecular and analytical techniques (*Raguso, 2008*). Floral scent not only attracts pollinators (*Klahre et al., 2011*), but also manipulates them through chemical mimicry (*Stökl et al., 2010*) and repels herbivores (*Junker and Blüthgen, 2010*), altogether increasing plant fitness (*Kessler et al., 2008*). However, research studying the function of floral scent has been divided along two themes with little cross-fertilization: 1) studies examining the fitness effects of floral scent without the causal behavioral responses of pollinators (*Kessler et al., 2008*), or 2) studies examining the sensory physiology of pollinators, neglecting the ecological consequences for the plant (*Raguso, 2008*). Here, we meld these approaches and show that floral scent increases the fitness of individual flowers not only by increasing their detectability (*Raguso and Willis, 2002*), but also by enhancing the pollinator's foraging motivation, and demonstrate that this is mediated by olfactory receptors on the tip of the moth's proboscis which detect floral scent.

The hawkmoth *Manduca sexta* (*Figure 1A*, *Video 1*) is a major pollinator of the wild tobacco *Nicotiana attenuata* in the Great Basin Desert (USA) (*Kessler et al., 2008*; *2010*; *2015*). *N. attenuata* emits a relatively simple floral scent dominated by a single compound: benzyl acetone (BA) (*Euler and Baldwin, 1996*). In spite of this simplicity, producing BA might come at considerable metabolic but especially ecological cost, as BA might attract not only nectar thieves and florivores, but also female hawkmoths in search of oviposition sites (*Baldwin et al., 1997*; *Kessler et al., 2010*). Plants might therefore reduce the amount of floral volatiles released as much as possible without losing their pollination services. Field experiments using plants in which the emissions of BA had been

silenced by RNAi of the biosynthetic gene *NaChal1* (CHAL) have shown that BA is required to maximize pollination success (*Kessler et al., 2008*; *2015*). It was suggested that lacking this scent made plants in nature 'invisible' to hawkmoth pollinators. However, the precise mechanisms by which odors of single flowers influence pollinator behavior and thereby plant fitness have rarely been examined in the direct interaction between plant and pollinator (*Klahre et al., 2011*; *Riffell et al., 2008*). Hence, how floral scent emitted by individual flowers functions in this mutualistic interaction remained unknown, particularly as it is unclear how pollinators detect single volatile compounds within complex natural environments (*Hansson and Stensmyr, 2011*; *Riffell et al., 2014*).

## Results and discussion

We investigated the function of floral scent in the context of individual flower-moth interactions, by offering individual male moths the choice between BA-emitting flowers (i.e. empty-vector transformed flowers (EV)) and non-emitting flowers (i.e. CHAL) in a free flight tent (24 m × 8 m × 4 m, 10 CHAL and 10 EV plants, spaced 50 cm apart). The flight tracking revealed that moths chose to visit the same number of emitting and non-emitting flowers in a random sequence (*Figure 1B*, *Figure 1— figure supplement 1A*, probability of changing between EV or CHAL flowers during consecutive visits: 0.47). In a second bioassay conducted in a wind tunnel (2.4 m × 0.9 m × 0.9 m, moonlight [0.5 lux of sunlight spectrum]), we presented plants with either emitting or non-emitting flowers to individual moths and analyzed their flight patterns, approaches and flower contacts. In none of these analyses, we found any difference between plants with emitting and non-emitting flowers (*Figure 1— figure supplement 1B,C*). These results suggest that visual cues and general vegetative plant odors provided sufficient information for the moths to locate flowers, consistent with previous work using artificial flowers (*Raguso and Willis, 2002*) and clearly showing that non-scenting flowers are not 'invisible' to moths.

If non-scenting flowers are found by moths, why is plant fitness reduced? Does BA emission change the pollination probability? To test this, we loaded the moth's proboscis with a standardized number of pollen grains using a fine brush. When such pollen-enhanced moths were allowed to forage freely on antherectomized *N. attenuata* flowers, seeds produced per flower of EV and CHAL plants differed significantly (*Figure 1C*). Scentless flowers matured very few seeds, reflecting the inferior pollination services provided by the moths despite a similar number of visits. This result highlights the importance of BA emission for the fitness of individual flowers and confirms the results of previous studies, which investigated the effects of BA emission on plant fitness at a population level (*Kessler et al., 2008*; *2015*). But if the flowers were equally detectable by the moths, what behavioral mechanism was responsible for the plant fitness consequences?

To analyze the effect of BA emission on moth behavior in greater detail, we quantified the time invested by a moth at individual flowers in a wind tunnel assay. The moths spent significantly more time at emitting than at non-emitting flowers (*Figure 2A*) particularly while trying to insert their proboscis, so even before tasting the floral nectar (*Figure 2—figure supplement 1A*). However, having successfully inserted their proboscis, the time of nectar uptake was similar between them (*Figure 2— figure supplement 1B*). This suggests that BA emission increased the motivation of moths to forage when individual flowers were evaluated at a close range, possibly because BA emissions, are closely linked to the physiological state and thereby also to the potential nectar amount of a flower (*Bhattacharya and Baldwin, 2012*; *Yon et al., 2015*; *Kessler et al., 2015*). By increasing the probing time in BA-emitting flowers, moths increased their success rate at their first as well as at consecutive flower visits and, therefore, collected more nectar per flower visit in tent (*Figure 2B*, *Videos 1* and *2*, *Figure 2—figure supplement 1C*) and wind tunnel assays (*Figure 2—figure supplement 1D*). These results agree with a study using different *Petunia* lines which found that although flower scent aided navigation, increased nectaring was the most consistent effect of floral scent (*Klahre et al., 2011*).

The large fitness consequences of floral volatiles for both moth and plant beg the question: how do moths evaluate the headspace of individual flowers? The wide spread of the antennae and their distance from the flower resulting from the moth's long proboscis which is fully extended during nectaring suggests that the olfactory spatial resolution of the antennae might be too low to resolve individual flowers in an inflorescence or even between neighboring plants (*Willis et al., 2013*). Hence, we inferred that the moth's proboscis might play a role in flower perception (*Goyret and*

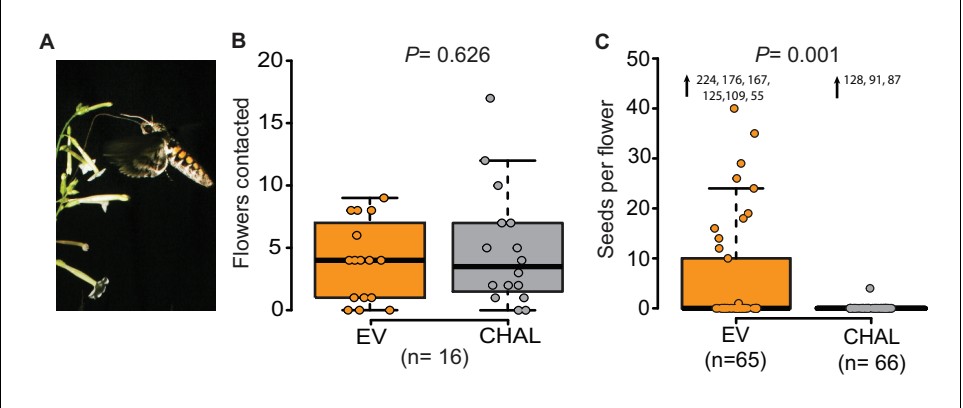

**Figure 1.** Even though *M. sexta* visited the same number of benzyl acetone (BA)-emitting (EV) and non-emitting flowers (CHAL), BA-emitting flowers received superior pollination services, increasing seed production. (**A**) *M. sexta* feeding from *N. attenuata* flowers. (**B**) Number of EV and CHAL flowers visited per moth on each foraging flight when 10 randomly placed plants per line were presented in a two-choice, free-flight tent assay (Wilcoxon signed rank test). (**C**) Seeds matured per antherectomized flower after visitations by a moth experimentally loaded with pollen (Wilcoxon rank sum test). Extreme values are shown as numbers.
The following figure supplement is available for figure 1:

**Figure supplement 1.** Free flight response of *M. sexta* towards flowers emitting and not emitting BA.

---

*Raguso, 2006*). Using reverse transcription PCR, we qualified the accumulation of transcripts of olfactory genes in the proboscis of *M. sexta*. Similar to the mosquito *Aedes aegypti*, but contrasting with predictions for nectar feeding insects (*Jung et al., 2015*), we found that the olfactory co-receptor Orco was expressed in the tip region of the proboscis along with the ionotropic co-receptor, IR25a (*Figure 3A*). Notably, Orco was only expressed in the first centimeter of the proboscis whereas the ionotropic co-receptor, IR8a, was only found in upper sections. This heterogeneous distribution of olfactory genes is consistent with the idea that the moth proboscis plays a more complex role in chemoreception than previously thought (*Reiter et al., 2015*). Screening the proboscis tip by scanning electron microscopy (*Figure 3B*, *Figure 3—figure supplement 1A,B*) we found a sensillum type that was not previously described for *M. sexta* (*Reiter et al., 2015*). This sensillum resembled the known sensillum styloconicum, but instead of a single tip pore, had a multiporous cone (*Figure 3B$_4$*). Similar sensillum types have been described in other lepidopteran species (*Faucheux, 2013*), but their function remained unknown, although the presence of odorant-binding proteins suggested a role in olfaction (*Nagnan-Le Meillour et al., 2000*). We used an antibody raised against Orco (Nolte et al., unpublished), and found a single Orco- positive cell only in the first multiporous sensilla styloconica (mSt) at the tip of the proboscis (*Figure 3C*, *Figure 3—figure supplement 1*).

Given the presence of potential olfactory sensilla on the proboscis of *M. sexta*, we wondered whether neurons housed in these sensilla play a role in the detection of BA and could thus help explain the pollination differences of EV and CHAL flowers. We performed single sensillum recordings and tested the response of neurons present in all sensillum types occurring at the tip of the proboscis to an air puff of BA at an ecologically relevant concentration (0.1 mM *Kessler and Baldwin, 2007*). Only neurons in the Orco-positive sensillum reacted to this compound (*Figure 4B*, *Figure 4—figure supplement 1A*). In a further test with 41 other

**Video 1.** *M. sexta* foraging on EV flowers emitting BA in a free-flight tent.

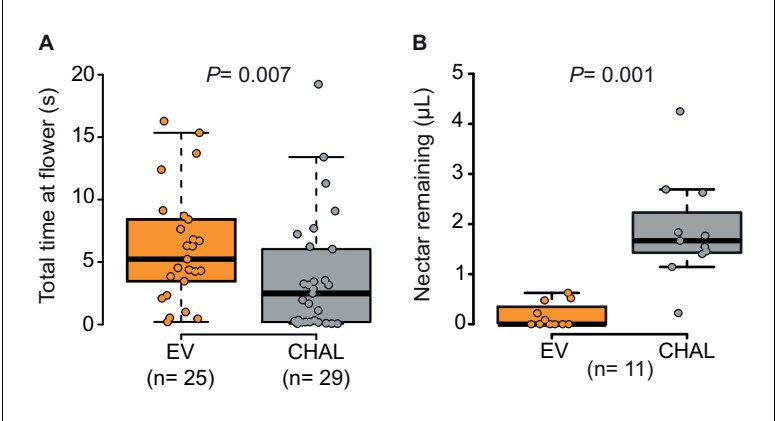

**Figure 2.** Moths spent more time and removed more nectar from BA-emitting (EV) flowers than from scentless (CHAL) flowers. (**A**) Time spent by moths at single flowers (Wilcoxon rank sum test) in a wind tunnel assay. (**B**) Nectar remaining in flowers after moths attempted to feed in a two choice tent assay (Wilcoxon signed rank test).

The following figure supplement is available for figure 2:

**Figure supplement 1.** Behavior of *M. sexta* while foraging on EV and CHAL.

ecologically relevant odorants, the first mSt was found to be more sensitive to BA and the structurally related benzylacetate (*Figure 4C*, *Figure 4C—source data 1*). Though these results show that neurons in the proboscis tip of *M. sexta* can detect volatile BA, it remained unclear whether the moth would also respond behaviorally to this compound based only on the input from neurons of the proboscis sensilla.

To disentangle the proboscis input from other chemosensory organs, we devised a behavioral experiment in which the corolla tube of a flower was replaced by a Y-maze choice assay (*Figure 4D*). Each arm of the Y-maze was either connected to a source of humidified air or humidified air scented with BA (0.1 mM). By drawing air directly behind the entrance of the Y-maze, the experimental set-up excluded antenna-based olfaction. Hence, as soon as a moth entered the flower aperture during free hovering flight, only the proboscis experienced the air stream containing either a solvent control, or BA (*Video 3*). During their first and subsequent insertions, the moths chose both Y-tube arms with equal frequency (*Figure 4F*, *Figure 4—figure supplement 1B*), but inserted their proboscis for a significantly longer time into the arm containing the BA-scented air (*Figure 4E*), demonstrating that the moth was able to detect BA with only the proboscis. Moths seem to use the olfactory input from the proboscis not for orientation on the corolla, but rather to assess the specific quality of an individual flower, consistent with the notion that the close-range orientation of the proboscis on the flower can be informed by mechanical and visual cues (*Goyret and Raguso, 2006*; *Sponberg et al., 2015*).

Our findings show that hawkmoths are well adapted to visit and detect volatiles of single flowers. Floral volatiles, such as BA, not only function as navigational cues (*Haverkamp et al., 2016*), but also inform pollinators about the identity and the physiological state of individual flowers (*Bhattacharya and Baldwin, 2012*; *Yon et al., 2015*). Only BA-emitting flowers encourage the moth to visit a flower long enough to lead to successful pollination. Our results show that floral scent is an essential chemical feature for hawkmoths to gain nectar from, and pollinate, a single flower. Interestingly, many flowers require that their pollinators

**Video 2.** *M. sexta* attempting to forage on Chal flowers not emitting BA in a free-flight tent.

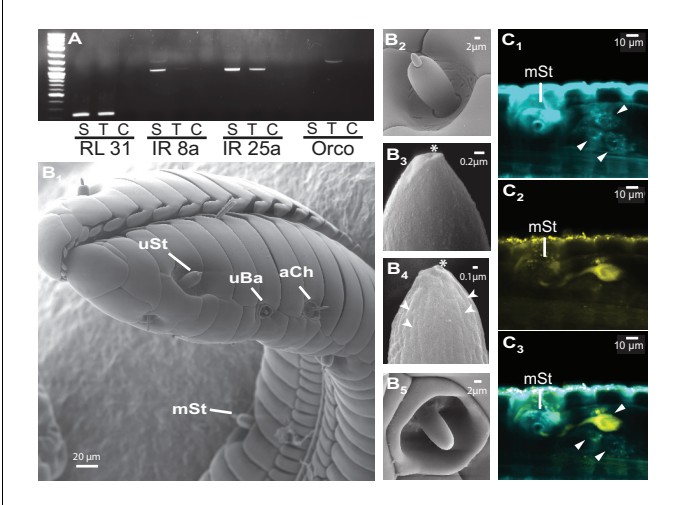

**Figure 3.** The *M. sexta* proboscis harbors sensilla, which house sensory neurons expressing olfactory genes. (**A**) Reverse transcription PCR using either the proboscis shaft (S), the first centimeter of the proboscis tip (T) or a water control (C) with primers for the three major olfactory co-receptors (IR8a, IR25a, Orco). The transcripts of the ribosomal gene RL131 and water were included as positive and negative control. (**B1**) Scanning electron microscopy images of the *M. sexta* proboscis tip show three types of potential chemosensory sensilla: sensilla styloconica (**B2**) with a uniporous (uSt (**B3**)) and multiporous (mSt (**B4**)) cone and uniporous sensilla basiconica (uBa (B5, *Figure 3—figure supplement 1B*)) as well as aporous sensilla chaetica (aCh, *Figure 3—figure supplement 1A*). Asterisks mark tip pore, arrowheads indicate side pores. Neuronal labeling using anti-bodies against horseradish peroxidase (**C1**) and against Orco(**C2**) indicate three neurons close to the first mSt sensillum (arrows), of which one expresses the olfactory co-receptor Orco (**C3**, *Figure 3—figure supplement 1*).
The following figure supplement is available for figure 3:

**Figure supplement 1.** Different sensillum types were found in tip region of the *M. sexta* proboscis, but only multiporous sensilla styloconica (mSt) were determined to be Orco positive.

acquire particular handling skills on their first visits, before the insects are able to use the flower efficiently (*Laverty, 1994*). This energy investment by the pollinator not only helps ensure outcrossing for the plant, but provides the insect with a more exclusive nectar source (*Heinrich, 1979*). In a recent study, inexperienced *M. sexta* were found to sometimes expend more energy on handling flowers than they gained from the nectar; if additional experience increases foraging efficiency, this would compel the moth to visit additional flowers of the same species (*Haverkamp et al., 2016*). Notably, the ability to smell BA with the tip of the proboscis may not only increase the motivation of *M. sexta* to invest energy into BA emitting flowers, but also strengthen the moth's learning of these flowers, as the nectar reward becomes associated with the presence of BA. Such a BA-conditioned learning rate might help to ensure a positive energy balance for the moth while at the same time ensuring cross-pollination for the plant (*Heinrich and Raven, 1972*). However, the question to what extent the interaction of moths and plants relies on the moths' learning ability requires additional attention in future studies.

Although both – metabolically costly and risky in terms of herbivory – thousands of plant species actively emit floral scent (*Wright and Schiestl, 2009*). These emissions might be a consequence of the physiological requirement for scent compounds by certain pollinator guilds when collecting nectar, even when visual cues would be sufficient to attract pollinators to a plant. The ongoing evolutionary interaction between plants and pollinators relies heavily on floral scent (*Parachnowitsch et al., 2012*; *Schiestl and Johnson, 2013*), and may explain the absence of scent-free plants in native *N. attenuata* populations (*Kessler et al., 2015*). To pollinators with the appropriate sensory system, floral scent provides a wealth of information, highlighting the importance of chemical communication in this mutualism, on which many of our crops rely (*Radera et al., 2015*).

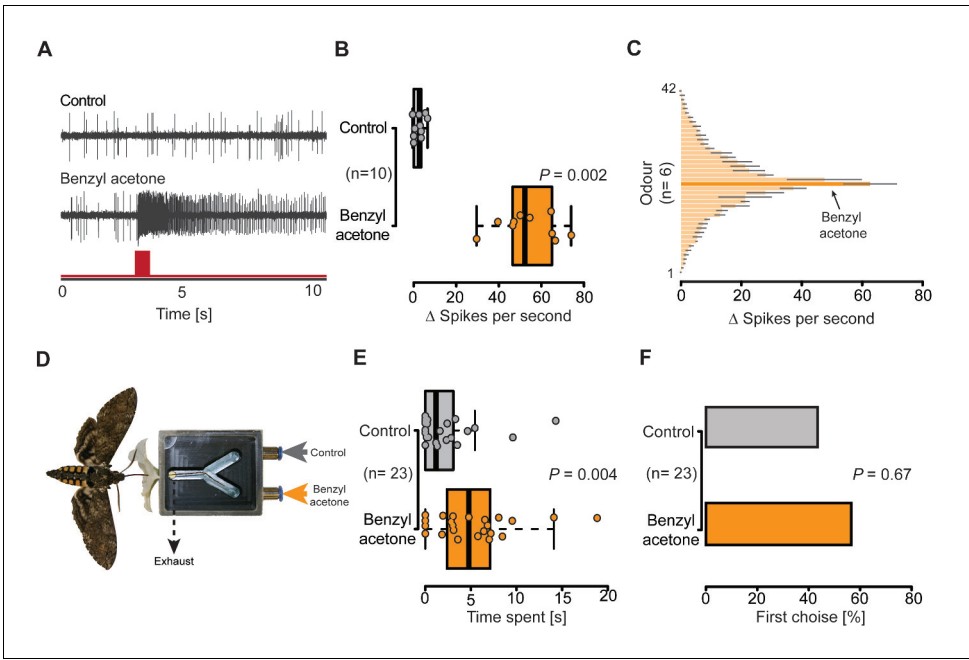

**Figure 4.** Olfactory sensory neurons housed in proboscis sensilla respond to BA and are sufficient for flower evaluation. (A) Single sensillum recordings from the first multiporous sensilla styloconica. Upper trace depicts a characteristic response to the water control; lower trace shows a response to BA from the neuron in the same sensillum. Red bar indicates time of stimulus. (B) Boxplot shows Δ spikes per second recorded from the first mSt when stimulating with water control or BA (0.1 mM) for 0.5 s. Neurons responded with a significantly higher spike rate to BA than to the water control (Wilcoxon signed-rank test). (C) Response profile of the first mSt to 42 different odorants. Black bars indicate S.E.M. Names and spike rate of each odorant can be found in *Figure 4C— source data 1*. (D) Behavioral assay to test the response to either humidified air with BA (0.1 mM) or humidified air only. Exhaust excludes antennal olfactory input. (E) Moth inserted their proboscis significantly longer into the arm in which BA was present (Wilcoxon signed-rank test). (F) Moths chose equally often between Y-tube arms containing BA-scented air or solvent control at the first approach (Exact binominal test).

The following source data and figure supplement are available for figure 4:

**Source data 1.** Response spectrum of the first multpourus sensillum to ecological relevant odors.

**Figure supplement 1.** BA does not influence all types of sensilla on the proboscis of *M. sexta* and does also not enable the moth to navigate actively towards BA.

# Materials and methods

## Plants

We used two transgenic *Nicotiana attenuata* Torr. (Solanaceae) lines derived from *Agrobacterium tumefaciens* (strain LBA 4404) transformation of wild type *N. attenuata* plants which were collected in a native population at the DI Ranch (Santa Clara, UT, USA) in 1988 and subsequently inbred for 22 generations (*Krügel et al., 2002*). Both lines have been described earlier, empty vector control plants (EV) transformed with pSOL3NC (line number A-04-266-3) (*Bubner et al., 2006*), as well and plants silenced by RNAi in the production of floral scent, CHAL line (*N. attenuata* chalcone synthase; pRESC5CHAL, line number A-07-283-5) (*Kessler et al., 2008*). Seeds were sterilized and germinated on Petri dishes with Gamborg's B5 media as described in *Krügel et al. (2002)*. Petri dishes with 30 seeds were maintained under LD (16 hr light and 8 hr dark) conditions in a growth chamber (Percival, Perry, Iowa, USA) for 10 days, and seedlings were transferred to small pots (TEKU JP 3050 104 pots, Pöppelmann, Germany) with Klasmann plug soil (Klasmann-Deilmann, Germany) in the glasshouse. After 10 days, plants were transferred to 9 cm × 9 cm pots for wind tunnel experiments or to 1 l pots for tent experiments.

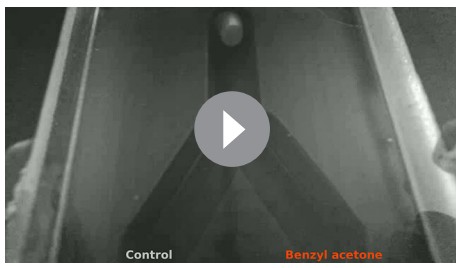

**Video 3.** Y-maze of the proboscis choice assay, BA-scented air is provided at the right arm and the solvent control in the left arm. Odors were removed by applying a vacuum at the entrance of the Y-maze.

The glasshouse growth conditions are described in *Krügel et al. (2002)*. For use in the wind tunnel, plants were transferred to a York Chamber (Johnston Controls, USA) with an inverted day/night cycle; daylight time was from 17–9, maintained at 25°C and night time temperatures were 22°C, with a humidity of 60–80%. Plants used in the tent experiment were cultivated in the Max Planck Institute for Chemical Ecology (MPICOE), Jena, Germany. After attaining the rosette stage of growth, plants were transferred to a second glasshouse, maintained at the same conditions, located in Isserstedt, Germany, where the plants were cultivated until used for the tent experiments.

## Insects

Moths used in the wind tunnel experiments were obtained from a colony maintained at the in Jena, Germany. Animals were reared as previously described (*Koenig et al., 2015*). Eggs were collected from female *M. sexta* moths, which could freely oviposit on *N. attenuata* plants. For the tent experiments, eggs from native *M. sexta* populations at the Utah field station were collected and shipped to Germany. After hatching, caterpillars were maintained on artificial diet (wind tunnel) or on *Nicotiana tabacum* plants (tent) at an ambient temperature of 27°C, 70% relative humidity and a light regime of 16:8 (light: dark). Fifth instar caterpillars were transferred into individual wood chambers for pupation. One week before hatching, pupae were sexed and male pupae were transferred to a flight cage with 15.5 hr daylight and 7.5 hr dim light (0.5 lux). Temperature and relative humidity were set to 25°C and 70% during day-light phase and to 20°C and 60% during the dim light phase. A transition phase of 30 min was used between phases. Animals were used for experiments 72–76 hr after hatching.

## Wind tunnel

No-choice assays were performed in a Plexiglas wind tunnel (220 cm × 90 cm × 90 cm). Charcoal-filtered air was pushed through the tunnel at a speed of 0.37 m/s. Air temperature and relative humidity was adjusted to 25°C and 70%. Plants were transferred to the wind tunnel chamber at least 1 hr before the experiment; to avoid contamination, plants were cultivated in a separated compartment with an additional charcoal air filter. Directly before each trial, a single plant was placed at the front of the wind tunnel with the flower positioned 70 cm above the tunnel floor, 20 cm from the tunnel front and 45 cm from each tunnel side.

Moths were placed individually in mesh-cages (15 cm × Ø13 cm) 1 hr before the experiment and transferred to a pre-exposure chamber set to the same light and climate conditions as the wind tunnel. For each trial, a single *M. sexta* moth was placed on a platform 35 cm above the tunnel floor, 20 cm from the tunnel end and 45 cm from each tunnel side. After placement, every animal was given 5 min to innate wing- fanning. Animals which did not start wing-fanning (63% ) within this time frame were considered as non-responders and excluded from subsequent analyses. After take-off, the behavior of each moth was recorded for 4 min using a custom-made 3D video tracking system. The tracking system consisted of four cameras (Logitech C615, USA, infrared filter removed) recording at 30 Hz and a resolution of 800 × 600 pixels (each pixel 0.3 cm$^2$). Using a background subtraction algorithm implemented in C, the 3D position of the moth was calculated at a rate of 10 Hz. Based on these tracking data, we analyzed the flight pattern of the moth during the last two seconds before encountering the flower in the wind tunnel using costume-written Matlab scripts (Mathworks, USA). All recorded flight tracks were cross-checked with video data and only complete recordings were used for further analyses. In order to avoid learning effects, we only considered the first flower approached by each moth.

Moth behavior at the flower was recorded at a rate of 30 Hz by a fifth camera (Logitech C615, infrared filter removed), which had been placed into the wind tunnel at a distance of about 30 cm

from the flower. Recordings were automatically started by a custom-written movement detection algorithm. Flower probing times and total contact time were measured by manually analyzing the individual video files. Similar to the flight analyses, we only considered the first flower contact in the statistical analyses.

## Tent

To emulate a natural environment, we conducted pollination experiments in a large tent (height, 4 m; width, 8 m; length, 24 m) with an enclosed roof to protect from rain and lateral mesh for natural airflow (*Kessler et al., 2015*). Plants were moved from the glasshouse to the tent which was located directly adjacent to the glasshouse. Experiments in the tent were conducted between August 27 and September 8 2014. Ten plants of each of the two lines EV and CHAL were aligned in rows at the central section of the tent. EV and CHAL plants were positioned directly next to each other, even touching inflorescences. The position of the EV and CHAL plants were changed after each moth, to minimize potential position effects. In order to use only freshly opened flowers, all open flowers were removed each morning, before experiments were conducted. Six to eight male *M. sexta* moths were released sequentially per night. A new moth was released only when the previous had stopped flying and every moth was only used once. Single flower visitations were observed, the genotype and time at a single flower was noted, and after the visitation, the approached flowers were removed to measure the remaining nectar. Each moth had up to 10 flower encounters and for each moth the mean nectar gain across all flower encounters was calculated.

## Nectar and pollen analysis

Directly after each experimental wind tunnel trial, both the moth and the plant were removed. The remaining nectar in the flower was measured by carefully removing the flower base and removing the nectar with a pre-weighed capillary. The nectar amount was determined by reweighing the capillary and subtracting the two weights. In the tent trials, nectar volume was quantified directly using a BLAUDBRAND graduated capillary with a volume of 25 μL (Brand, Germany) by gently removing the corolla (*Kessler et al., 2007*).

The pollen load on the moth proboscis was determined by rinsing the proboscis three times in 1 mL of 1% Tween solution. 10 μL of 0.5% safranin (Sigma Aldrich, Germany) were added to each sample to stain the pollen outer layer. The samples were vortexed and centrifuged for 2 min at 10000 rpm. Thereafter, the supernatant was discarded without disturbing the pollen pellet and 100 μL distilled water was added to each sample. The samples were then vortexed and 10 μL were pipetted into a four-field Neubauer-counting chamber to determine the pollen number in each sample. Every sample was counted twice independently, and the mean value was used for statistical analysis.

## Cross pollination experiment

To measure pollination rates in EV and CHAL plants by *M. sexta* in the wind tunnel, fully developed flowers were emasculated in the previous corresponding daylight morning cycle to avoid self-pollination (*Kessler et al., 2008*). For this, 3–4 flowers per plant of each line (EV and CHAL) were used in the wind tunnel, one plant and one moth at each time. Fresh pollen was collected in the corresponding morning from plants not being used for pollination. EV pollen which had been collected the previous night was rubbed on the hawkmoth proboscis using a fine brush prior to its release in the wind tunnel, in order to measure the pollen delivery to experimental flowers. If moths did not take flight voluntarily within 3 min after being placed in the wind tunnel they were excluded from the study. Moths, which took flight (73% ), were allowed to do so for four minutes in each wind tunnel trial. The numbers of matured capsules, as well as the seeds produced from each capsule were counted after ripening. Capsules were collected shortly before opening, approximately 14 days after the experiment, dried in a desiccator, and once opened, the seeds were counted in petri dishes. After each trial the pollen from each *M. sexta* was collected by washing the proboscis to ensure that similar amounts of pollen had been placed on the proboscis. For pollen counts, the same procedure as for the pollen retrieval was used. On average, we found that 548.75 (n= 41, SEM= 88.85) pollen grains had been placed on a single proboscis. No difference was found between moths tested with EV (n= 21, mean= 569, SEM= 92.3) or CHAL (n=20, mean= 500, SEM= 159.3) plants (Student's t-test, p= 0.71).

## Scanning electron microscopy

*M. sexta* proboscises were cut 1 cm from the tip and fixed in 4% glutaraldehyde at 4°C overnight. Proboscises were then dehydrated in an ascending ethanol series (70%, 80%, 90%, 96%, 3x 100% ethanol, 10 min each), critical point dried (BAL-TEC CPD 030, Bal-Tec Union Ltd., Liechtenstein), mounted on aluminium stubs with conductive carbon cement (Agar Scientific, UK) and sputter coated with gold on a BAL-TEC SCD005 (Bal-Tec, Liechtenstein). Specimens were examined in a LEO 1530 Gemini scanning electron microscope (Zeiss, Germany) set at 8 kV and 11 to 15 mm working distance.

## Immunohistochemistry and confocal laser scanning microscopy

The tip region of 20 *M. sexta* proboscises were carefully dissected into three small parts, cutting behind the first, before the fourth and behind the fifth sensillum styloconica. Directly after dissection, the proboscis parts were fixed in 4% paraformaldehyde (ROTH, Germany) in 1 M NaHCO3 (Sigma Aldrich, pH 9.5) overnight at 4°C. Subsequently, the samples were washed six times for 30 min in 1× phosphate-buffered saline containing 0.1% Trition X (PBS-T) (Sigma Aldrich, USA) and thereafter blocked for 3 hr in normal goat serum (NGS). The primary anti-body against Orco (kindly provided by Prof. Jürgen Krieger, University of Halle-Wittenberg, Germany) was applied at a 1:500 dilution in 2% NGS- PBS-T and incubated for 5 days at 4°C. Detection of the Orco antibody was performed by incubating in a goat-anti rabbit antibody linked to Alexa 488 (Invitrogen, USA) at a dilution of 1:200 in 2% NGS-PBS-T for 3 days at 4°C. In addition, we added an goat anti-horseradish peroxidase antibodies conjugated to Cy3 (Jackson Immuno Research, USA) at a dilution of 1:50 in 2% NGS-PBS-T to visualize neuronal tissue. For visualization, the samples were mounted in 50% glycerol on a microscope slide and scanned using confocal laser scanning microscopy (LSM 880, Zeiss, Germany). Alexa 488 was exited using the 488 nm line of the microscopes Argon laser, while a Helium Neon 543 laser was used to activate Cy3. Signals were detected by a spectral detector (quasar: 490–553 nm and 555–681 nm). All pictures were taken using a 20× air objective (N.A. 0.8). Scanning resolution was set to 1024 × 1024 pixel.

## Total RNA isolation

Proboscises of ten male *M. sexta* were dissected and were cut 1 cm from the tip. Each tissue sample (tips and rest) was directly transferred to Tri-reagent (Sigma-Aldrich, USA). The samples were then homogenized with two 3 mm steel beads (Qiagen, Germany) using a TissueLyser (Qiagen, Germany) for 5 min at 50 Hz. Samples were stored at -20°C. Finally, RNA isolation was performed using TRI-Reagent (Sigma-Aldrich, USA) according to the manufacturer's instructions.

## cDNA synthesis

RNA samples were treated with TurboDNAse (Ambion, USA) according to the manufacturer's instructions. DNAse was removed using Tri-reagent following the instructions of the producer. RNA was dissolved in 25 μL RNA storage solution (Ambion, USA). For cDNA synthesis 1 μg total RNA per sample was used as template for the Super Script III kit (Invitrogen, Canada).

## Reverse transcription-PCR

For RT-PCR dNTPS (Thermo Fisher Scientific, Lithuania), cDNA, gene-specific primers and the Avantage 2 Polymerase mix (Clontech, Canada) were used following the manufacturer's instructions. Primers were designed according to Koenig et al. (2015): (RL31: GGA GAG AGG AAA GGC AAA TC and CGG AAG GGG ACA TTT CTG AC; MsexIR8a: CAA CCC CGA CGC GTA TCC GTA TCC and TTA CGG CCT ATA TTC ATT TTT AGG AAA AAC GCT TAT ATA TG; MsexIR25a: GGA GTC CGT ATA GCT ATC AGA ATA ATC GAG and TCA AAA TTT AGG TTT CAA ATT AGA TAA ACC TAA ATT TCT GGA TC; MsexORCo: ATG ATG GCC AAA GTG AAA ACA CAG G and CTA TTT CAG CTG CAC CAA CAC CAT G). Reaction was done in a thermocycler (GeneAmp PCR System 9700, PE Applied Biosystems, USA) with 95°C for 1 min, followed by 35 cycles of 95°C for 30 s, 60°C (for MsexIR25a: 62°C) for 30 s and 68°C for 90 s. The final step was incubation at 68°C for 3 min. The samples were loaded on a 1.5% agarose gel.

## Electrophysiology

For electrophysiological recordings, moths were placed into a 15 mL reaction tube, from which the tip had been cut; in such a way that only the proboscis would extend from the tube. Eachanimal was then mounted on a microscope slide, and the proboscis was fixed with dental wax. Next we unrolled the first centimetre of the proboscis and fixed this part upside down on a small wax pedestal, so that most of the sensilla were approachable for electrophysiological recordings. Subsequently, the preparation was positioned under a microscope (BX51W1, Olympus, Japan) and a tungsten reference electrode was inserted into the proboscis shaft. The recording electrode was then inserted into the target styloconic sensillum via a motorized, piezo-translator-equipped micromanipulator (DC-3K/PM-10, Märzhauser, Germany). A constant air stream of humidified air was applied to the preparation. For stimulus delivery either 0.1 mg BA diluted in distilled water or distilled water only was loaded onto a filter paper, inserted in a glass pipette and puffed onto the proboscis using a Syntech stimulus controller (CS- 55, Syntech, The Netherlands). For the odor screen individual compounds were diluted in hexane ($10^{-2}$ v/v) and 10 μL were loaded on to a filter paper and puffed as described before. A single puff lasted for 0.5 s. The recorded signal was then amplified (UN-06, Syntech, The Netherlands), digitally converted (IDAC-4, Syntech, The Netherlands), and recorded at a rate of 2400 Hz using AutoSpike v3.2. (Syntech, The Netherlands). Traces were exported as ASCII files and manually analyzed using R. Spikes were counted 2 s before the stimulus onset and 2 s thereafter. The number of spikes before the stimulus was then subtracted from the spikes counted after the stimulus onset. The resulting number of Δ spikes was then divided by the number of seconds analyzed. In all experiments, three day old male moths were used.

## Proboscis choice

Olfactory preference of the proboscis was tested in a custom-built Y-maze (5 cm × 3 cm × 0.5 cm). Previous studies had found a 0.1 mM suspension of BA in nectar to be ecologically relevant in the interaction between *M. sexta* and *N. attenuata* (*Kessler and Baldwin, 2007*). Here, we tested 10 μL of 0.1 mM BA suspension in distilled water against the same amount of distilled water only. Both stimuli were pipetted onto a small filter paper discs and placed into 50 mL glass bottles. Bottles were connected to the Y-tube arms via Teflon tubing (Ø 6 mm). Charcoal-filtered air was pushed into the bottles so that the air flow at each Y-tube arm reached 0.1 L/ min. To prevent the moths' antenna from contacting BA headspace and assure a homogenous flow, air was removed from the opening of the Y-tube at a rate of 0.2 L/ min. The movement of the moth proboscis was recorded via a video camera (Logitech C615, infrared filter removed) at 30 Hz. Videos were captured using the software package Media recorder (Noldus, The Netherlands) and subsequently viewed and manually analyzed using EthoVision (Noldus, The Netherlands). For tests, the Y-maze set-up was placed into the wind tunnel described above and moths were allowed to forage freely for 4 min. In order to attract the moths to the Y-maze, we attached the corolla of a freshly cut *Nicotiana alata* flower, which does not release BA (*Raguso et al., 2003*), onto the Y-maze opening.

## Acknowledgements

We thank Sandor Nietsche, the EMZ Jena, and Lydia Gruber for their help with the SEM images, Jürgen Krieger for providing the Orco anti-body, Tamara Spingler for supporting the Y-tube assays and Daniel Veit for engineering the set-ups for behavior experiments. Funding was provided by the European Research Council advanced grant no. 293926 to ITB and the Max-Planck-Society.

## Additional information

### Competing interests

ITB: Senior editor, *eLife*. BSH: Vice President of the Max Planck Society, one of the three founding funders of eLife, and a member of eLife's Board of Directors. The other authors declare that no competing interests exist.

## Funding

| Funder | Grant reference number | Author |
| --- | --- | --- |
| European Research Council | advanced grant no. 293926 to Ian T. Baldwin | Ian T Baldwin Danny Kessler |

The funders had no role in study design, data collection and interpretation, or the decision to submit the work for publication.

## Author contributions

AH, FY, DK, Conception and design, Acquisition of data, Analysis and interpretation of data, Drafting or revising the article; IWK, CM, CK, Acquisition of data, Analysis and interpretation of data, Drafting or revising the article; BSH, ITB, MK, Conception and design, Analysis and interpretation of data, Drafting or revising the article

## Author ORCIDs

Alexander Haverkamp, http://orcid.org/0000-0003-3512-9659
Ian T Baldwin, http://orcid.org/0000-0001-5371-2974
Markus Knaden, http://orcid.org/0000-0002-6710-1071
Danny Kessler, http://orcid.org/0000-0003-0410-116X

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
