## [Decision Letter]

Thank you for submitting your article "Hawkmoths evaluate scenting flowers with the tips of their tongues" for consideration by *eLife*. Your article has been reviewed by 2 peer reviewers, and the evaluation has been overseen by Marcel Dicke as the Reviewing Editor and Detlef Weigel as the Senior Editor.

The following individual involved in review of your submission has agreed to reveal their identity: Rob Raguso (peer reviewer).

The reviewers have discussed the reviews with one another and the Reviewing Editor has drafted this decision to help you prepare a revised submission.

This manuscript describes an interesting study of the perception of benzyl acetone by the proboscis of naive *Manduca sexta* female moths. The description of the multi-pored sensillum that can respond to the floral volatile benzyl acetone is new and electrophysiological as well as behavioural assays show that the sensillum responds to benzyl acetone. Yet, the reviewers feel that some more information is needed on this new sensillum to better understand its role in the perception of the flowers. The main issues to be dealt with are the following: (a) the title where the word proboscis would be the correct word to be used, (b) the specificity of the sensillum by testing more than only the response to benzyl acetone, but also other volatiles emitted by *Nicotiana attenuata*. After all, *N. attenuata* nectar produces more volatiles than only benzyl acetone (Kessler et al. 2007 Plant J.), (c) is the sensillum also involved in sugar perception: does the sensillum also house a sugar responding dendrite? and (d) the role of benzyl acetone perception by the probiscis of naïve moths versus moths that have gained experience that do not require scented nectar; this has consequences for the discussion (Discussion section, last paragraph) on the herbivore-pollinator trade-off. In addition, the reviewers have made many valuable additional comments that will be useful to further improve the manuscript.

Reviewer #1:

This manuscript builds upon previous studies dissecting the role of floral chemistry in mediating interactions between Nicotiana attenuata plants and their insect visitors, including Manduca sexta as a co-pollinator/herbivore. This study addresses a neglected dimension of flower visitation by M. sexta, the role of the proboscis during the final stages of flower visitation, where a moth physically negotiates a nectar tube or spur and discovers nectar.

The authors ask an intriguing question: could the extended proboscis detect volatiles at the scale of an individual flower, and if so, could that assist in their handling of those flowers, and might such handling have fitness consequences for the plant?

What they find is fascinating. Indeed, the proboscis has a (surprise!) multi-pored sensillum that can respond to a floral volatile (benzyl acetone = BA). They record from it in response to BA, show that ORCO is expressed within the sensillum (as one would expect in an antenna) and show with a Y-tube assay that the presence of BA extends moth probing time in the correct arm of the "nectary". I looked for such sensilla using SEM 20 years ago, but apparently not carefully enough, as it looks superficially just like the single pored styloconic sensillum type known to respond to sugar solutions in this and other hawkmoths. Kelber, A. (2003). Sugar preferences and feeding strategies in the hawkmoth *Macroglossum stellatarum*. Journal of Comparative Physiology A, 189(8), 661-666.

The authors use a series of flight cage, wind tunnel and y-tube assays to test for the effects of scent on different spatial scales, exploiting a line of silenced *N. attenuata* plants (CHAL) that do not emit floral BA, and comparing them with the empty vector (EV) control plants.

The primary result, consistent across experiments, is that scent perception by the extended proboscis is associated with extended probing time in a scented flower, and that this in turn is associated with better acquisition of nectar and better pollen transfer. One of several caveats (see below, details) of this study is that the assays were performed with flower naïve moths, so we don't yet know if these behaviors persist, or whether an experienced moth has acquired the motor skills in handling flowers that obviate the need for them to be scented. Certainly, *M. sexta* can learn to visit many, many kinds of natural and artificial flowers that lack scented nectar or even strong corolla scent, if they are immersed in a scent cloud. I have several comments below that address areas where the authors should clarify their wording or interpretation of their results.

1) Introduction: The first sentence is weak; ecosystem function really is about nutrient and energy flow and not about pollination success. It is just window dressing and can be cut.

2) Results and Discussion: "possibly because BA emissions, unlike visual cues, are a good predictor of nectar reward". This sentence requires clearer explanation. It refers specifically to the *N. attenuata* system but cites a global study by Junker and Bluthgen. Presumably both BA and visual cues are imprecise indicators of nectar if the flowers have been emptied recently? A cue emanating from the nectar itself, either nectar-specific scent (nicotine?) or relative humidity, would be the most accurate index of nectar presence, as discussed in several other papers (e.g. van Arx et al. PNAS).

3) Results and Discussion, third paragraph: precision of wording. It is generally regarded as negative for foraging animals to increase handling time in an optimal foraging sense, because it reduces profitability of a resource by increasing the denominator (reward over HT). What you mean here, I think, is that scented flowers increase persistence of probing beyond mechanosensory stimulation, and this leads to greater success in flower discovery by untrained moths. I would recommend calling this variable "probing time" instead of handling time to avoid confounding these ideas, which are very different. In fact, beyond what is shown in Figure 2, one would like to see a pair of acquisition curves for moths learning to use EV vs. those learning to use CHAL. Have the authors done this?

4) Concerning the multiporous cone sensillum found near tip of proboscis; does it also house a sugar responding dendrite? Have you tried recording from it when both BA and sucrose are present? Similarly, is there any way to cauderize or KO the tongue tip? Zinc sulfate dip treatments, as have been done with crickets responding to cuticular HCs? The problem is that the same or similar sensilla probably respond to sugar.

5) Y-maze for proboscis, blind test of whether it can orient to source of scent. Figure 4 nicely shows that moths spend more time with proboscis in scented arm (vs. humidified), but do not ask if they do same for humidified air over dry air. Also, the authors should show a similar panel for first choice, even if there is no difference. That would provide a nice parallel to the wind tunnel experiment showing no differences in flower choice but significant increase in probing time. (Oh, I see that this is shown Figure 4—figure supplement 1 – it might be more effective if moved to Figure 4).

6) Figure 4 caption, part B is written as if aqueous solutions with and without BA are being presented in the Y tube arm, but the description in the text is of presenting humidified air with vs. without BA. Please clarify here, as is more clear in the Methods.

7) Results and Discussion: the following lines are misleading: "Floral volatiles, such as BA, not only function as navigational cues (Raguso and Willis, 2002), but also inform pollinators about the state of individual flowers". That is not quite accurate.

The array assays in this paper and the one cited (Raguso and Willis 2002) clearly show that *M. sexta* moths do not choose individual flowers, in flight, by whether they are scented or not (p. 691, section entitled Lack of Discrimination between Natural and Paper Flowers), if those flowers are embedded in a scent cloud. That finding is consistent across nearly all published experiments with *M. sexta*. Raguso and Willis 2002 was not a wind tunnel test, but a set of bioassays performed in small flight cages in still air. Thus, scent was not interpreted as a navigational aid, but rather as a sign stimulus, synergizing visual display to elicit proboscis extension and flower visitation.

Instead, what the authors' data show is that BA encourages extended probing time once a moth has decided to probe a flower (chance, in their behavioral assays). They do not have an experimental result here that indicates remote (without probing) evaluation of flowers with vs. without nectar.

Once *M. sexta* and other hawkmoths commit to feeding in a patch of flowers, they extend the proboscis and keep it extended until they leave that patch. That observation led me and others to suspect that the extended proboscis could function as a third antenna, so to speak, providing spatial information to the brain about odor plume sources. To my knowledge, that possibility has not been examined, and it seemed unlikely to me given the moths' hierarchical choices for probing at bright objects once stimulated by scent (Raguso and Willis 2005, Goyret et al. 2007).

8) I would add that the last paragraph of the Results and Discussion is not accurate either, given the large body of work showing how floral scent impacts foraging behavior in flower naïve *M. sexta* that have not yet learned to associate scent with nectar, and many papers about innate preferences for scent (see Schiestl 2015 New Phytologist for a recent review). The present experiments were not done with experienced moths, so it is possible that scent-aided probing is primarily a benefit to flower-naïve moths, rather than being generally essential for them to choose and pollinate flowers.

9) Methods, subsection “Insects”: female moths.

10) Methods, subsection “Tent”: spatial resolution of scent cloud: if the EV and CHAL plants are touching, it is likely that the entire array is embedded within a scent cloud that precludes moths distinguishing between odor sources on the fly. If they repeated these experiments with plants separated by a few meters, they might get a different result. This problem is common to all array experiments performed with hawkmoths over the last decade (using *Petunia, Mimulus, Hemerocallis*).

11) Methods, subsection “Tent”: please clarify whether batches of moths were released individually or in batches. The latter would violate assumptions of independence, or would render each released batch as a replicate, as in *Drosophila* bioassays.

12) Methods, subsection “Cross pollination experiment”: Please address the extent to which adding pollen to tongue tips impacted moth performance. They don't like being handled and having their probosces extended, but it is possible that doing this during the photophase, when they are quiescent, would not impair foraging behavior during scotophase.

13) Methods, subsection “Scanning electron microscopy”: provide replicates please, for males and females.

14) Methods, subsection “Proboscis choice”: wouldn't it be more accurate to have used sucrose solutions with BA instead of BA in distilled water? The colligative properties of the sucrose (salting out) likely would change emission rates of BA from that solution.

15) Figure 1—figure supplement 1: the diagram does not show an array (i.e. a checkerboard), but rather, two lanes of plants, color coded to indicate EV vs. CHAL. Is that true, or is the diagram a simplification? Please clarify, because the former design is more effective than the latter in terms of avoiding side of cage biases, etc.

16) Panel C does not make sense. I don't understand what you are showing on the left side (Y axis wind tunnel width, x axis wind tunnel length) – please explain more clearly what you are testing here and what you found vs. expectations. You are not experimentally varying the dimensions of the wind tunnel! The right panel simply shows a non-significant trend, with 3:2 ratio of sample sizes, and is not strongly supported.

Reviewer #2:

This study makes an interesting contribution to our understanding of an important problem. It shows that certain sensilla on the proboscis of *Manduca* are olfactory. A very clever behavioral paradigm is used to show that these sensilla detect a major component, benzyl acetone (BA), of the scent of wild *Nicotiana attenuata*. The study also shows that flowers engineered not to emit BA receive inferior pollination services, and that less nectar is removed from them. The simplest interpretation of these results taken together is that pollination and nectar removal depend on the sensilla, although this is not demonstrated directly by ablation or masking experiments.

An extensive study of the specificity of the sensilla is beyond the scope of this study, but the story would be more compelling if it were shown that the sensilla showed at least some odorant-specificity.

---

## [Author Response]

This manuscript describes an interesting study of the perception of benzyl acetone by the proboscis of naive Manduca sexta female moths. The description of the multi-pored sensillum that can respond to the floral volatile benzyl acetone is new and electrophysiological as well as behavioural assays show that the sensillum responds to benzyl acetone. Yet, the reviewers feel that some more information is needed on this new sensillum to better understand its role in the perception of the flowers. The main issues to be dealt with are the following: (a) the title where the word proboscis would be the correct word to be used, (b) the specificity of the sensillum by testing more than only the response to benzyl acetone, but also other volatiles emitted by Nicotiana attenuata. After all, N. attenuata nectar produces more volatiles than only benzyl acetone (Kessler et al. 2007 Plant J.), (c) is the sensillum also involved in sugar perception: does the sensillum also house a sugar responding dendrite? and (d) the role of benzyl acetone perception by the probiscis of naïve moths versus moths that have gained experience that do not require scented nectar; this has consequences for the discussion (Discussion section, last paragraph) on the herbivore-pollinator trade-off. In addition, the reviewers have made many valuable additional comments that will be useful to further improve the manuscript.

For the title of the manuscript we would now suggest: “Hawkmoths evaluate scenting flowers with the tip of their proboscis”.

We followed the reviewers’ advice and have now tested the electrophysiological response of the multiporouse sensillum styloconica to 41 additional, ecological relevant odors to further characterize the specificity of the sensillum. We have included these results as an additional panel in Figure 4, as a table in [Supplementary-material SD1-data] and in the fifth paragraph of the Results and Discussion section.

The question whether this sensillum also houses a sugar responsive neuron, is certainly of interest for the foraging ecology of *M. sexta*. However, we feel that it is beyond the scope of our study as we are mainly focusing here on the influence of floral volatiles detected by the moth proboscis. Furthermore, our behavioral data indicates that BA influences the decision of the moth before the moth contacts the nectar, but not necessarily during feeding (Figure 2—figure supplement 1). We have now aimed to make this clearer to the reader by rephrasing the text in the third paragraph of the Results and Discussion section.

Similarly, we feel that the influence of BA on the flower learning of *M. sexta* is an interesting and important point, which would deserve a more thorough investigation than we can achieve in this study. Nonetheless, we would hypothesize that experience might further increase the difference between emitting and non-emitting flowers as moth that forage on emitting flowers have a higher change of receiving a nectar reward and might therefore continue to forage on emitting flowers, and not at non-emitting once. Moths are also more likely to learn a flower which provides both visual as well as olfactory information (Riffell and Alarcón, 2013), which might again increase the preference of the moth for scented flowers. This hypothesis is consistent with the results of our tent experiments in which we allowed the moth to encounter up to 10 flowers consecutively and analysed the mean nectar gain of each moth (we have now tried to clarify this in the Methods section, subsection “Tent”). We found that after these consecutive flower encounters, the difference in the nectar uptake between the emitting and the non-emitting flowers was even greater than in our wind tunnel assays, in which the moth encountered only a single flower. Moreover, we have analysed the percentage of foraging success for consecutive visits on EV and CHAL plants in the free flight tent and included this as Figure 2—figure supplement 1. These results indicate the high success rate of *M. sexta* on emitting flowers is indeed maintained also during consecutive flower visits whereas the foraging success on non-emitting flowers remains relatively low. Hence we would expect that the difference between emitting and non-emitting flowers would rather be maintained or even increased (as one could assume that moths become more efficient in handling emitting flowers, after they learned to associate a nectar reward with the presence of BA). We have added a short discussion on this point in the seventh paragraph of the Results and Discussion section.

Finally, we would like to point out that our study only was conducted with male moths (we tried make this clearer in the first paragraph of the Results and Discussion section and in the Methods subsection “Tent”) and we have therefore removed the discussion on the plant-herbivore trade-off to avoid any further confusion; the interaction between oviposition and nectar foraging has been and will be an important question for future research, and you were correct in pointing out that by including it here, the waters are only muddied.

Reviewer #1:

This manuscript builds upon previous studies dissecting the role of floral chemistry in mediating interactions between Nicotiana attenuata plants and their insect visitors, including Manduca sexta as a co-pollinator/herbivore. This study addresses a neglected dimension of flower visitation by M. sexta, the role of the proboscis during the final stages of flower visitation, where a moth physically negotiates a nectar tube or spur and discovers nectar.

[…]

The primary result, consistent across experiments, is that scent perception by the extended proboscis is associated with extended probing time in a scented flower, and that this in turn is associated with better acquisition of nectar and better pollen transfer. One of several caveats (see below, details) of this study is that the assays were performed with flower naïve moths, so we don't yet know if these behaviors persist, or whether an experienced moth has acquired the motor skills in handling flowers that obviate the need for them to be scented. Certainly, M. sexta can learn to visit many, many kinds of natural and artificial flowers that lack scented nectar or even strong corolla scent, if they are immersed in a scent cloud. I have several comments below that address areas where the authors should clarify their wording or interpretation of their results.

We fully agree that flower learning is an important dimension in the interaction which deserves more attention than we can provide in this study. However, we would also like to point out that in our flight tent experiment, the moths were allowed to visit up to 10 different flowers in a row and a mean value of the remaining nectar was then calculated over these 10 flowers (Figure 2). Therefore, if the observed effect would have only been present during the first flower encounter one would have expected that the effect would be diminished after consecutive flower visits; however, we found the opposite. The effect was even stronger after these consecutive visits than what was observed in the wind tunnel where the moth had only excess to a single flower (Figure 2—figure supplement 1). Furthermore, we have now calculated the success rate of the moth in the tent assay over consecutive flower visits and found that both the high success rate on emitting flowers as well as the relatively low success rate on non-emitting flowers are maintained over consecutive flower encounters (Figure 2—figure supplement 1). Thus learning did not appear to change the overall success rate of the moth. Nonetheless, we would hypothesize that the moth might still be able to learn how to handle flowers more effectively and we would expect that in the case of the scented flower this effect might be even stronger than with non-scented flower, since the moths were more likely to obtain a reward from the scented flowers and reward quantity is thought to influence the strength of the motor learning (Wright et al., 2009). Moreover, the fact that the moths can also make use of visual and olfactory information, the scent might further reinforce and increase their learning rate (Riffell and Alarcón, 2013). We have added a short discussion on this in the seventh paragraph of the Results and Discussion section.

1) Introduction: The first sentence is weak; ecosystem function really is about nutrient and energy flow and not about pollination success. It is just window dressing and can be cut.

We agree with the reviewer on this point and have modified the sentence to “Floral scent has been associated with insect pollination since the 18^th^ century (Sprengler, 1793); however, only recently have the complex functions of floral volatiles been investigated in detail, due to the availability of new molecular and analytical techniques (Raguso, 2008).”

2) Results and Discussion: "possibly because BA emissions, unlike visual cues, are a good predictor of nectar reward". This sentence requires clearer explanation. It refers specifically to the N. attenuata system but cites a global study by Junker and Bluthgen. Presumably both BA and visual cues are imprecise indicators of nectar if the flowers have been emptied recently? A cue emanating from the nectar itself, either nectar-specific scent (nicotine?) or relative humidity, would be the most accurate index of nectar presence, as discussed in several other papers (e.g. van Arx et al. PNAS).

The reviewer is right to point out that BA emissions from the corolla are not directly linked to the nectar production and we have rephrased the sentence in the third paragraph of the Results and Discussion section to be more precise.

While BA is not a direct predictor of floral nectar it remains a good indicator of the physiological state and the metabolic activity of *N. attenuata* flowers (Bhattacharya and Baldwin, 2012, Yon et al., 2015; Kessler et al., 2015) and might thus help to inform moths seeking more rewarding flowers. We furthermore agree with the reviewer that other cues such as humidity and potentially nicotine might be more informative to *M. sexta* in selecting flowers with standing nectar, and the detection of these cues might be interesting questions for further studies.

3) Results and Discussion, third paragraph: precision of wording. It is generally regarded as negative for foraging animals to increase handling time in an optimal foraging sense, because it reduces profitability of a resource by increasing the denominator (reward over HT). What you mean here, I think, is that scented flowers increase persistence of probing beyond mechanosensory stimulation, and this leads to greater success in flower discovery by untrained moths. I would recommend calling this variable "probing time" instead of handling time to avoid confounding these ideas, which are very different. In fact, beyond what is shown in Figure 2, one would like to see a pair of acquisition curves for moths learning to use EV vs. those learning to use CHAL. Have the authors done this?

We would like to thank the reviewer for this comment. We have exchanged “handling time” with “probing time”. Learning curves have not been done, since the focus of this study was on the ecological characterization of a newly discovered sensillum working at a single flower basis rather than the differential learning capacity for scented and non-scented flowers. Nevertheless, we agree that learning and/ or performance curves of *M. sexta* foraging on scented and unscented flowers would be an important further step to understand the interaction between *M. sexta* and *N. attenuata*. We have now included the success rate of *M. sexta* after consecutive flower visits (Figure 2—figure supplement 1).

4) Concerning the multiporous cone sensillum found near tip of proboscis; does it also house a sugar responding dendrite? Have you tried recording from it when both BA and sucrose are present? Similarly, is there any way to cauderize or KO the tongue tip? Zinc sulfate dip treatments, as have been done with crickets responding to cuticular HCs? The problem is that the same or similar sensilla probably respond to sugar.

We agree with the reviewer that cauterizing or ablating the sensilla or even the proboscis tip is problematic as these sensilla are likely to also house additional mechanosensory or gustatory neurons which might also influence the flower probing behavior of *M. sexta*. A double test with sugar has not been done, given that the experimental procedure to measure volatile and soluble cues are different, therefore both potential responses are difficult to measure simultaneously. Moreover, the objective of the study was more focused on olfaction than on taste. As our Y-maze assay shows, the presence of volatiles alone is sufficient to increase the moths probing time. Whether or not gustatory cues like sugar add to this effect, is interesting, but beyond the scope of this study.

5) Y-maze for proboscis, blind test of whether it can orient to source of scent. Figure 4 nicely shows that moths spend more time with proboscis in scented arm (vs. humidified), but do not ask if they do same for humidified air over dry air. Also, the authors should show a similar panel for first choice, even if there is no difference. That would provide a nice parallel to the wind tunnel experiment showing no differences in flower choice but significant increase in probing time. (Oh, I see that this is shown Figure 4—figure supplement 1 – it might be more effective if moved to Figure 4).

We would like to thank the reviewer for raising these points and agree that it might be more effective to move Figure 4—figure supplement 1 to Figure 4, which we have now done.

We have always analyed the response towards 10 µL of a 0.1 mM BA suspension in distilled water on filter paper again 10 µL water on filter paper. Hence we have always tested the response of the proboscis in a slightly humidified environment. Given that BA is mostly present in a humid environment, both in the nectar and on the corolla (von Arx et al., 2012), we believe that this is the most likely situation in which the proboscis would encounter BA. As water was present on both arms of the Y-maze (see comment on correct figure legend below), humidity is less likely to explain the observed results of increased probing time in the presence of BA.

6) Figure 4 caption, part B is written as if aqueous solutions with and without BA are being presented in the Y tube arm, but the description in the text is of presenting humidified air with vs. without BA. Please clarify here, as is more clear in the Methods.

Legend in Figure 4 has been modified.

7) Results and Discussion: the following lines are misleading: "Floral volatiles, such as BA, not only function as navigational cues (Raguso and Willis, 2002), but also inform pollinators about the state of individual flowers". That is not quite accurate.

The array assays in this paper and the one cited (Raguso and Willis 2002) clearly show that M. sexta moths do not choose individual flowers, in flight, by whether they are scented or not (p. 691, section entitled Lack of Discrimination between Natural and Paper Flowers), if those flowers are embedded in a scent cloud. That finding is consistent across nearly all published experiments with M. sexta. Raguso and Willis 2002 was not a wind tunnel test, but a set of bioassays performed in small flight cages in still air. Thus, scent was not interpreted as a navigational aid, but rather as a sign stimulus, synergizing visual display to elicit proboscis extension and flower visitation.

Instead, what the authors' data show is that BA encourages extended probing time once a moth has decided to probe a flower (chance, in their behavioral assays). They do not have an experimental result here that indicates remote (without probing) evaluation of flowers with vs. without nectar.

Once M. sexta and other hawkmoths commit to feeding in a patch of flowers, they extend the proboscis and keep it extended until they leave that patch. That observation led me and others to suspect that the extended proboscis could function as a third antenna, so to speak, providing spatial information to the brain about odor plume sources. To my knowledge, that possibility has not been examined, and it seemed unlikely to me given the moths' hierarchical choices for probing at bright objects once stimulated by scent (Raguso and Willis 2005, Goyret et al. 2007).

This is a valid point. We have recently investigated the flight of *M. sexta* towards different *Nicotiana* flowers in the absence of meaningful visual cues in more detail, and found that in such a situation the hawkmoths are also able to navigate towards an odor source emitting the scent of *N. attenuata* (Haverkamp et al., 2016). We added this citation to the main text and rephrased the second part of the sentence to “…,but also inform pollinators about the identity and the physiological state of individual flowers.” and added Bhattacharya and Baldwin (2012) and (Yon et al., 2015) as an additional references in line 203.

In the sentence we point at the general assumption that flower scent, such as BA, work over distance as navigational cues (Haverkamp et al. 2016). The reviewer is of cause right to point out that flower volatiles are not directly indicating whether nectar is present in the flower or not. Nevertheless, the volatile emissions correlate with the physiological activity of the flower (Bhattacharya and Baldwin, 2012), which might still be an important cue for the moth as freshly opened, young scenting flowers are more likely to contain nectar than older non-scenting flowers.

The hypothesis of the reviewer, that the proboscis might be used as a ‘third antenna’ to obtain more spatial information on the odor source is certainly an interesting one, however just like the reviewer, we also do not think that this is very likely. In our experiments we have not observed any indication, that the proboscis would enable an enhanced flower location; on the contrary in the Y-maze, we observed no difference in the first choice of the proboscis, which indicates that the proboscis is not able to navigate by chemical cues directly (Figure 4).

8) I would add that the last paragraph of the Results and Discussion is not accurate either, given the large body of work showing how floral scent impacts foraging behavior in flower naïve M. sexta that have not yet learned to associate scent with nectar, and many papers about innate preferences for scent (see Schiestl 2015 New Phytologist for a recent review). The present experiments were not done with experienced moths, so it is possible that scent-aided probing is primarily a benefit to flower-naïve moths, rather than being generally essential for them to choose and pollinate flowers.

We agree with the reviewer that the sentence has been phrased too strongly and we have changed this sentence and following paragraph accordingly in the seventh paragraph of the Results and Discussion section. As discussed above, the data are not consistent with the hypothesis that learning alters the outcome of the plant-pollinator interaction observed here. However, we expect that the difference between scented and non-scented flowers would remain similar or even increase further due to the ability of the moth to associate the floral scent with the nectar reward. In the tent assay the moths were allowed to probe 10 consecutive flowers and were therefore already after the first visit, not fully naïve. However, despite their experience, the observed effect was not only maintained but was even increased in comparison to the single-experience experiments in the wind tunnel (Figure 2 and Figure 2—figure supplement 1). Additionally, our new results also indicated that the moth maintained their high success rate on scented flowers as well as their relatively flow success rate on non-scented flowers also over several flower encounters (Figure 2—figure supplement 1).

9) Methods, subsection “Insects”: female moths.

Changed.

10) Methods, subsection “Tent”: spatial resolution of scent cloud: if the EV and CHAL plants are touching, it is likely that the entire array is embedded within a scent cloud that precludes moths distinguishing between odor sources on the fly. If they repeated these experiments with plants separated by a few meters, they might get a different result. This problem is common to all array experiments performed with hawkmoths over the last decade (using Petunia, Mimulus, Hemerocallis).

We are thankful to the reviewer for raising this issue. It would indeed be possible that non-emitting flowers are “hiding” in the scent cloud of the emitting plants, which could (Figure 2) and we had briefly discussed this in the fourth paragraph of the Results and Discussion section. We have now extended the relevant sentence.

Nonetheless, our results from the tent are in line with our findings in the wind tunnel where plants were introduced individually and non-scented flowers could thus not have profited from the scented flowers (Figure 1—figure supplement 1), indicating that visual cues and general plant odorants are sufficient for a moth to detect a non-scented flower.

11) Methods, subsection “Tent”: please clarify whether batches of moths were released individually or in batches. The latter would violate assumptions of independence, or would render each released batch as a replicate, as in Drosophila bioassays.

The reviewer raises an important point and we have tried to clarify this in our Methods section (subsection “Tent”). Moths were released sequentially and visually tracked in the tent until they stop flying; therefore, individual moths did not repeat the assay and at no time there was more than one active moth in the tent.

12) Methods, subsection “Cross pollination experiment”: Please address the extent to which adding pollen to tongue tips impacted moth performance. They don't like being handled and having their probosces extended, but it is possible that doing this during the photophase, when they are quiescent, would not impair foraging behavior during scotophase.

We agree with the reviewer that hawkmoths are sensitive to handling and we have thus only included animals which took flight voluntarily within 3 min (73%) after the pollen was applied. This information was indeed missing from the Methods section and we have now added this in the subsection “Cross pollination experiment”. However, we would further argue that moths approaching emitting and non- emitting flowers were handled in the same way, it is therefore conceivable that our treatment might have decreased the overall rates of pollinations, but not have influenced the observed dramatic differences between emitting and non-emitting flowers.

13) Methods, subsection “Scanning electron microscopy”: provide replicates please, for males and females.

The difference between the male and female hawkmoth proboscis is a very interesting aspect to explore as we also mention in the Discussion. Nonetheless, we feel that it is out of the scope of our current paper, which focuses exclusively on male moth and nectar foraging. We now state in the manuscript more clearly that we worked only with male moths.

14) Methods, subsection “Proboscis choice”: wouldn't it be more accurate to have used sucrose solutions with BA instead of BA in distilled water? The colligative properties of the sucrose (salting out) likely would change emission rates of BA from that solution.

The reviewer raises a very interesting point, especially as many hawkmoth pollinated flowers vary strongly in their sugar concentration and composition (Contreras et al., 2013; Kaczorowski et al., 2005). This might then also influence the perception of secondary nectar metabolites. However, in *N. attenuata* BA is mainly emitted at the corolla limb surface (Euler and Baldwin 1996; Kessler and Baldwin, 2007). We would therefore assume that the BA emissions which influence the behavior of *M. sexta* in our study are mostly independent of the nectar sugar concentration and composition.

However, we followed the reviewer’s advice and tested the neuronal response to a 0.1 mM BA solution solved in a 0.5 M sucrose solution in 4 animals, as we did not find any significant difference to the BA dissolved in water (median= 37.25 ∆spikes/s, p= 0.12), we prefer to exclude these results from the manuscript.

15) Figure 1—figure supplement 1: the diagram does not show an array (i.e. a checkerboard), but rather, two lanes of plants, color coded to indicate EV vs. CHAL. Is that true, or is the diagram a simplification? Please clarify, because the former design is more effective than the latter in terms of avoiding side of cage biases, etc.

The plants were arranged as in the diagram for feasible tracking and removal of visited flowers, and we have now aimed to make this clearer in the Materials and methods (subsection “Tent”) and in the figure legend of Figure 1—figure supplement 1. The reviewer is right that the design might be more susceptible to side bias and we have altered the sides of EV and CHAL plants between different moths and experimental days to reduce this effect. This point has indeed been missing in the Methods section and we have now added this to the aforementioned subsection and to the legend of Figure 1—figure supplement 1. Moreover, moth visited EV and CHAL plants (i.e. plants on the left and right side) in the same proportion and in a random sequence (probability of visiting a plant on the other side: 0.47, Figure 1—figure supplement 1).

16) Panel C does not make sense. I don't understand what you are showing on the left side (Y axis wind tunnel width, x axis wind tunnel length) – please explain more clearly what you are testing here and what you found vs. expectations. You are not experimentally varying the dimensions of the wind tunnel! The right panel simply shows a non-significant trend, with 3:2 ratio of sample sizes, and is not strongly supported.

We are thankful to the reviewer for this comment and we have updated the figure legend. The left side of panel C represents a top view projection of the 3 dimensional flight tracking from two example flights in the frontal section of the wind tunnel. We agree that the sample size of the experiment is uneven and that the replicate number is rather low. However, the results are still of interest even if further careful experiments might be needed to further explore the mechanisms by which plant odors aid the navigation of *M. sexta* towards a plant or flower in the presents of visual cues.

Reviewer #2:

This study makes an interesting contribution to our understanding of an important problem. It shows that certain sensilla on the proboscis of Manduca are olfactory. A very clever behavioral paradigm is used to show that these sensilla detect a major component, benzyl acetone (BA), of the scent of wild Nicotiana attenuata. The study also shows that flowers engineered not to emit BA receive inferior pollination services, and that less nectar is removed from them. The simplest interpretation of these results taken together is that pollination and nectar removal depend on the sensilla, although this is not demonstrated directly by ablation or masking experiments.

An extensive study of the specificity of the sensilla is beyond the scope of this study, but the story would be more compelling if it were shown that the sensilla showed at least some odorant-specificity.

We agree that ablation or masking experiment might provide even further support for our findings; however, the tested sensilla also potentially contain mechano- and other chemoreceptors which might be important for flower probing. Results from such experiments might therefore seriously confound different sensory modalities and be difficult to interpret, as also mentioned by the first reviewer.

We agree that testing the response of the first multiporouse sensilla might enhance the manuscript and we have therefore followed the reviewer’s advice and tested the neuronal response to 41 additional odors (Figure 4 and [Supplementary-material SD1-data]) and presented these results in the fifth paragraph of the Results and Discussion section. Interestingly, we found that while the senisillum responded most strongly to BA and the structurally related benzyl acetate, it could also detect some other potentially relevant compounds such as nicotine although at a much lower level.